# Associations Between Big Five Personality Traits and Burnout Among Secondary Physical Education Teachers in South Korea

**DOI:** 10.3390/bs15111499

**Published:** 2025-11-05

**Authors:** Seungwoo Choi, Sungki Park, Ansu Lee

**Affiliations:** Department of Physical Education, Kyungpook National University, Daegu 41566, Republic of Korea; choisw75@knu.ac.kr (S.C.); s9953011@knu.ac.kr (S.P.)

**Keywords:** personality, burnout, secondary physical education teacher, cross-sectional study

## Abstract

Burnout among physical education (PE) teachers has become an urgent issue due to the profession’s distinctive physical, emotional, and social demands. This cross-sectional study examined the relationships between the Big Five personality traits and occupational burnout among secondary PE teachers in South Korea (*N* = 240). Burnout was measured using the Maslach Burnout Inventory–Human Services Survey, and personality traits were assessed with the Big Five Inventory. Correlation and hierarchical multiple regression analyses were conducted to explore the associations between personality traits and the three burnout dimensions: emotional exhaustion, depersonalization, and personal accomplishment. Neuroticism was positively associated with emotional exhaustion and depersonalization, whereas extraversion, agreeableness, and conscientiousness were negatively related to these dimensions. Personal accomplishment was positively linked to extraversion, agreeableness, conscientiousness, and openness, and negatively linked to neuroticism. Regression analyses confirmed that neuroticism was the strongest predictor of emotional exhaustion, while extraversion and agreeableness buffered depersonalization. Openness showed a positive association with depersonalization, suggesting a possible person–environment misfit in structured PE contexts. These findings indicate that personality profiles provide valuable insight into burnout vulnerability among secondary PE teachers and underscore the importance of personality-informed strategies to promote emotional well-being and sustainable professional growth.

## 1. Introduction

Teachers in modern educational settings are increasingly confronted with excessive workloads, emotional labor, and shifting institutional demands, all of which contribute to occupational burnout. Burnout not only undermines teachers’ well-being and professional performance but also compromises the quality of education ([28]; [36]). Within this context, physical education (PE) teachers are considered a particularly vulnerable group because they bear unique physical, psychological, and social responsibilities beyond classroom instruction. PE teachers are accountable for student safety in high-risk activity environments, while also managing extracurricular sports programs, athletic teams, and health-related initiatives. These multifaceted responsibilities intensify both stress and emotional strain ([30]; [64]).

Recent studies further highlight that PE teachers experience heightened role ambiguity and identity stress. Unlike other academic subjects, PE is often perceived as “play” rather than rigorous instruction, which can undermine professional legitimacy and foster dissatisfaction ([20]; [52]). PE teachers are also expected to assume central roles in student guidance, discipline, and the promotion of school well-being, responsibilities that expanded in scope during and after the COVID-19 pandemic ([44]; [67]). These expectations intensify emotional labor demands, requiring PE teachers to regulate or suppress their genuine emotions, thereby accelerating emotional depletion and risk of burnout ([36]; [69]). Recent evidence from secondary PE teachers likewise documents intense emotional experiences, control beliefs, and on-the-job coping processes ([59]).

A growing body of research has demonstrated that personality traits significantly influence how individuals cope with emotional labor and burnout ([60]). The Big Five Personality Traits model—comprising neuroticism, extraversion, agreeableness, conscientiousness, and openness—provides a robust framework for examining these relationships. Neuroticism is strongly associated with emotional instability and vulnerability to stress, whereas conscientiousness and agreeableness are linked to resilience, self-regulation, and cooperative social interactions ([6]; [32]; [66]). Personality also shapes interpersonal relationships and perceived social support in everyday contexts ([65]), offering a plausible social pathway through which extraversion and agreeableness may buffer burnout. More recent empirical work and meta-analyses suggest that neuroticism consistently predicts higher levels of emotional exhaustion and depersonalization, while extraversion serves as a protective factor against burnout ([3]; [54]). Complementing teacher-focused work, evidence from South African university students also shows robust links between Big Five traits and burnout, suggesting that the pattern generalizes across populations ([46]). Nonetheless, the protective effects of agreeableness, conscientiousness, and openness appear to vary across occupational contexts, including teaching ([40]).

Background characteristics also shape how personality traits translate into burnout. Gender differences are well documented, with women more frequently reporting higher emotional exhaustion due to caregiving roles and gender-related expectations ([50]), while women also tend to score higher on agreeableness and neuroticism ([15]). The teaching level has been associated with distinct stressors: middle school teachers often face students’ emotional and developmental challenges, whereas high school teachers experience academic performance pressures ([57]; [61]). Educational background may influence both professional attitudes and burnout risk, with postgraduate teachers balancing advanced expertise alongside increased responsibilities ([12]). By contrast, evidence for gender, work assignment, or marital status effects is less consistent, with some studies reporting negligible differences roles ([14]; [38]; [48]).

Although numerous studies have linked personality traits to burnout, most have focused on general or counseling teachers, treating the Big Five traits as static predictors rather than dynamic components operating within multidimensional burnout processes. Such an approach limits the theoretical clarity on how specific dispositional tendencies relate to distinct burnout dimensions—emotional exhaustion, depersonalization, and reduced personal accomplishment—and how these associations are conditioned by domain-specific emotional labor and contextual demands. Given the unique pedagogical, physical, and psychosocial responsibilities of PE teachers, the personality–burnout linkage may manifest differently from other teaching domains.

Specifically, PE teachers’ exposure to high-risk activity environments and extracurricular responsibilities creates situational pressures that can amplify or attenuate the influence of certain personality traits ([30]; [64]). Teachers high in neuroticism may be more sensitive to the ever-present possibility of physical injury or disciplinary incidents, intensifying emotional exhaustion through heightened vigilance and anticipatory stress ([1]; [40]). Conversely, those high in conscientiousness may overcommit to ensuring safety and performance standards, which, when institutional resources are insufficient, can paradoxically increase depersonalization and self-blame ([42]). Similarly, extraverted and agreeable teachers may initially thrive in socially interactive and dynamic contexts, yet extended extracurricular workloads and blurred work–life boundaries may erode their emotional resilience over time ([36]; [52]). These complex person–environment interactions suggest that the distinctive burnout patterns observed among PE teachers are not merely replications of known trait–burnout associations but context-dependent outcomes shaped by the ecological demands of physical education settings. Such contextualized interpretation extends the conventional trait–burnout framework beyond static associations, emphasizing PE-specific ecological stress dynamics.

The present study therefore advances the theoretical framing by integrating the Big Five model with the emotional and motivational contexts of PE. By examining how each trait differentially contributes to the subdimensions of burnout, this work refines existing personality–burnout models from a multidimensional and context-sensitive perspective. Moreover, by considering background characteristics, such as gender, teaching level, and educational background, it provides a more comprehensive understanding of how personality shapes adaptive or maladaptive responses to job-related stress among secondary PE teachers.

To address these gaps, the present study investigates how the Big Five personality traits predict different dimensions of burnout among secondary PE teachers. Specifically, it examines which personal traits function as protective or risk factors in buffering the emotional exhaustion that arises from identity-related stress and high emotional labor demands. In addition, the study explores whether these personality–burnout relationships differ across background characteristics, such as gender, work assignment, teaching level, educational background, and marital status. By contextualizing personality–burnout dynamics within the specific stress ecology of PE instruction, the study aims to clarify how dispositional and situational factors jointly shape teachers’ well-being and professional sustainability.

## 2. Methods

### 2.1. Research Design and Hypotheses

Building on these objectives, the present study employed a cross-sectional correlational design to test theoretically grounded hypotheses regarding the relationships between the Big Five personality traits and the three dimensions of teacher burnout (emotional exhaustion, depersonalization, and personal accomplishment) among secondary PE teachers. Based on the Big Five model and prior research on stress regulation and emotional labor in teaching, the following hypotheses were proposed:

**H1.** 
*Extraversion, agreeableness, conscientiousness, and openness is negatively associated with emotional exhaustion and depersonalization and positively associated with personal accomplishment.*


**H2.** 
*Neuroticism is positively associated with emotional exhaustion and depersonalization and negatively associated with personal accomplishment.*


**H3.** 
*Openness may also show a positive relationship with depersonalization in highly structured PE contexts, reflecting a possible misfit between autonomy-seeking dispositions and institutional constraints.*


In addition, exploratory analyses were conducted to examine whether these personality–burnout relationships differ across background characteristics, such as gender, work assignment, teaching level, educational background, and marital status.

### 2.2. Procedure and Participants

This study was conducted in accordance with the principles of the Declaration of Helsinki and was approved by the Kyungpook National University Institutional Review Board (approval no. 2024-0441). Participants were secondary PE teachers employed in D Metropolitan City, South Korea. Data were collected through an online questionnaire administered between September and December 2024.

Recruitment was facilitated through the official blog of the regional secondary PE teachers’ association. Upon the research team’s request, the association administrator posted an invitation with a unique survey link (Google Forms) in the announcements section. Participation was entirely voluntary, with no direct solicitation or snowballing. Before starting the survey, all respondents provided digital informed consent and were assured of anonymity, confidentiality, and the right to withdraw at any time without penalty.

In total, 241 responses were received, of which 240 were deemed valid after data screening for completeness and response consistency (effective response rate = 99.6%). The final sample consisted of 198 males (82.5%) and 42 females (17.5%). Among them, 145 (60.4%) reported having non-PE-related duties and 95 (39.6%) reported having PE-related duties. A larger proportion taught at middle schools (61.3%) than high schools (38.8%). Regarding educational attainment, 150 teachers (62.5%) held an undergraduate degree and 90 (37.5%) held a postgraduate degree or higher degrees. Most participants were married (75.8%), while 24.2% were single. The demographic characteristics are summarized in Table 1.

### 2.3. Measurements

To meet the study objectives, we developed a questionnaire as the data collection instrument. It comprised three parts.

#### 2.3.1. Sociodemographic Questionnaire

The first part contained five items assessing demographic characteristics (gender, work assignment, teaching level, educational background, and marital status). These variables were included because prior research suggests that such characteristics may contribute to variations in burnout and personality traits between teachers.

#### 2.3.2. Dependent Variable—Burnout (MBI-HSS; [41])

The second part of the questionnaire measured teachers’ occupational burnout using the Maslach Burnout Inventory–Human Services Survey (MBI-HSS; [41]). The scale consists of 22 items covering three subdimensions: emotional exhaustion (9 items), depersonalization (5 items), and personal accomplishment (8 items). Items were rated on a 7-point Likert scale (0 = never, 6 = every day). For example, an emotional exhaustion item is “I feel emotionally drained from my work,” and a depersonalization item is “I feel I treat some students as if they were impersonal objects.” Higher scores on emotional exhaustion and depersonalization and lower scores on personal accomplishment indicate higher levels of burnout. The MBI-HSS has consistently demonstrated a stable three-factor structure and strong internal consistency in the developers’ work and subsequent manuals, with subscale reliabilities typically ≥0.70 in prior research. In this study, the Cronbach’s α coefficients were 0.868 for emotional exhaustion, 0.794 for depersonalization, and 0.887 for reduced personal accomplishment. Confirmatory factor analysis (CFA) supported convergent validity, with all construct reliability (CR) values above 0.70. Although some average variance extracted (AVE) values were slightly below 0.50, they were acceptable according to [21]’s ([21]) criteria given the CR values were above 0.60.

#### 2.3.3. Independent Variable—Personality (BFI; [25])

The third part of the questionnaire assessed personality traits using the Big Five Inventory (BFI; [25]). The instrument consists of 44 items measuring five subdimensions: extraversion (8 items), agreeableness (9 items), conscientiousness (9 items), neuroticism (8 items), and openness (10 items). Items were rated on a 5-point Likert scale (1 = strongly disagree, 5 = strongly agree). For instance, an extraversion item is “I see myself as someone who is talkative,” and a conscientiousness item is “I see myself as someone who does a thorough job.” The BFI has been widely validated by its developers and subsequent research, demonstrating robust factorial validity and internal consistency across diverse populations ([25]; [26]; [63]). In this study, Cronbach’s α coefficients were 0.721 for extraversion, 0.651 for agreeableness, 0.822 for conscientiousness, 0.794 for neuroticism, and 0.855 for openness. CFA results supported construct reliability, with all CR values above 0.70. Although some AVE values were slightly below 0.50, the convergent validity was deemed acceptable following Fornell and Larcker’s criteria. While the internal consistency for agreeableness was slightly lower than the conventional 0.70 threshold, similar results have been reported in previous studies employing the BFI in the study by [6] ([6]) and across multiple populations in a recent reliability generalization meta-analysis ([23]). These findings indicate that moderate reliability for agreeableness is common and theoretically acceptable within the broader BFI literature.

### 2.4. Data Analysis

All statistical analyses were conducted using SPSS 29.0 and AMOS 27.0 software. Descriptive statistics were used to examine item- and factor-level characteristics of the burnout and personality measures. Normality was assessed via skewness and kurtosis, and internal consistency was evaluated using Cronbach’s α. To assess the scale reliability and validity, a confirmatory factor analysis (CFA) was estimated in AMOS using maximum likelihood estimation. The construct reliability (CR) and average variance extracted (AVE) were calculated to evaluate the convergent validity.

Pearson’s correlation analyses were performed to examine relationships between the study variables and independent-samples *t*-tests were conducted to examine the differences in personality traits and burnout according to the demographic characteristics.

To test the hypothesized relationships, hierarchical multiple regression analyses were conducted with each burnout dimension as the dependent variable. In the first step, relevant background variables were entered as control variables, followed by the five personality traits in the second step. For each model, standardized regression coefficients (*β*), 95% confidence intervals (CIs), and changes in explained variance (Δ*R*^2^) were calculated and examined for each model. Statistical significance was set at *p* < 0.05 for all analyses.

## 3. Results

### 3.1. Descriptive Statistics

Descriptive statistics for burnout and personality traits are presented in Table 2. Among the burnout subdimensions, reduced personal accomplishment scored the highest (M = 4.01), followed by emotional exhaustion (M = 2.26) and depersonalization (M = 1.34). For the personality traits, agreeableness (M = 3.89) was the highest, followed by conscientiousness (M = 3.83), openness (M = 3.35), extraversion (M = 3.27), and neuroticism (M = 2.54), indicating that overall, positive personality traits scored higher than negative traits. Skewness and kurtosis values for all variables were within the acceptable range (±2), confirming the assumption of normality for subsequent parametric analyses.

### 3.2. Correlation Analysis

The Pearson correlation coefficients between burnout and personality subdimensions are reported in Table 3. Emotional exhaustion and depersonalization were negatively correlated with extraversion, agreeableness, and conscientiousness (r = −0.218 to −0.374, *p* < 0.05), but positively correlated with neuroticism (r = 0.366 to 0.443, *p* < 0.05). In contrast, personal accomplishment was positively associated with extraversion (r = 0.417, *p* < 0.05), agreeableness (r = 0.440, *p* < 0.05), conscientiousness (r = 0.463, *p* < 0.05), and openness (r = 0.401, *p* < 0.05), while negatively associated with neuroticism (r = −0.432, *p* < 0.05). These findings indicate that positive personality traits tend to buffer against burnout, whereas neuroticism operates as a risk factor.

### 3.3. Exploratory Analyses of Differences in Personality and Burnout According to Background Variables

To examine the differences in personality and burnout according to background variables, independent-samples *t*-tests were conducted; the results are presented in Table 4.

First, by teaching level, middle school PE teachers reported higher depersonalization than high school PE teachers (t = 3.233, *p* < 0.05), whereas high school PE teachers reported higher personal accomplishment (t = −1.989, *p* < 0.05). Emotional exhaustion did not differ by teaching level (*p* > 0.05). Regarding personality traits, a significant difference was observed only in openness (t = −2.526, *p* < 0.05), with high school PE teachers scoring higher than middle school PE teachers.

Second, by educational background, teachers with undergraduate degrees reported higher depersonalization than those with postgraduate degrees (t = 1.751, *p* < 0.05), whereas postgraduate teachers reported higher personal accomplishment (t = −2.683, *p* < 0.05). Significant differences also emerged in extraversion, agreeableness, conscientiousness, and openness (all with *p* < 0.05), with postgraduate teachers scoring higher on these traits.

Third, no statistically significant differences emerged by gender (male vs. female), primary work assignment (PE-related vs. non-PE-related), and marital status (married vs. single) on any burnout dimension (EE, DP, PA) or Big Five personality trait (all with *p* > 0.05). Descriptively, the group means were closely aligned across categories, suggesting that any between-group differences, if present, are likely small in magnitude in this sample. This pattern is broadly consistent with prior reports of mixed or negligible differences by gender, role assignment, and marital status in teacher samples (e.g., [14]; [38]; [48]).

### 3.4. Effects of Personality Traits on Burnout

To test Hypotheses 1–3, hierarchical multiple regression analyses were conducted. The teaching level and educational background were entered as control variables in the first step, followed by the five personality traits in the second step. The control variables were not statistically significant in any of the models. The results are presented below.

#### 3.4.1. Hierarchical Multiple Regression Analysis Predicting Emotional Exhaustion from Personality Traits

Consistent with H2, neuroticism was positively associated with emotional exhaustion (*β* = 0.344, *p* < 0.05, 95% CI [0.332, 0.834]), whereas extraversion was negatively associated (*β* = −0.134, *p* < 0.05, 95% CI [−0.480, −0.006]). Other personality traits were not significant predictors. The overall model was significant (*F* = 9.760, *p* < 0.05, *R*^2^ = 0.227, Δ*R*^2^ = 0.219), indicating that the model accounted for about 22.7% of the variance in emotional exhaustion and that personality traits explained an additional 21.9% beyond the control variables. These findings highlight neuroticism as the most influential predictor of exhaustion, with extraversion serving as a modest protective factor. The results are presented in Table 5.

#### 3.4.2. Hierarchical Multiple Regression Analysis Predicting Depersonalization from Personality Traits

In line with H1–H3, extraversion (*β* = −0.170, *p* < 0.05, 95% CI [−0.542, −0.076]) and agreeableness (*β* = −0.261, *p* < 0.05, 95% CI [−0.970, −0.256]) were negatively associated with depersonalization, whereas neuroticism (*β* = 0.158, *p* < 0.05, 95% CI [0.021, 0.515]) and openness (*β* = 0.217, *p* < 0.05, 95% CI [0.135, 0.581]) were positively associated. The positive coefficient for openness supports H3, suggesting that autonomy-seeking tendencies may heighten depersonalization within highly structured PE contexts. Conscientiousness was not a significant predictor. The overall model was significant (*F* = 11.250, *p* < 0.05, *R*^2^ = 0.253, Δ*R*^2^ = 0.209), indicating that the model explained about 25.3% of the variance in depersonalization and that personality traits accounted for an additional 20.9% beyond the control variables. These results emphasize that extraversion and agreeableness may buffer against depersonalization, whereas neuroticism and openness increase the risk of depersonalization among secondary PE teachers. The results are presented in Table 6.

#### 3.4.3. Hierarchical Multiple Regression Analysis Predicting Personal Accomplishment from Personality Traits

Supporting H1–H3, extraversion (*β* = 0.217, *p* < 0.05, 95% CI [0.188, 0.638]), agreeableness (*β* = 0.155, *p* < 0.05, 95% CI [0.038, 0.724]), conscientiousness (*β* = 0.193, *p* < 0.05, 95% CI [0.109, 0.595]), and openness (*β* = 0.121, *p* < 0.05, 95% CI [−0.005, 0.423]) were positively associated with personal accomplishment, whereas neuroticism was negatively associated (*β* = −0.133, *p* < 0.05, 95% CI [−0.475, 0.003]). The modest positive association of openness with personal accomplishment is partially consistent with H3, suggesting that openness may function as a psychological resource under supportive conditions but as a vulnerability when constrained by rigid structures. The overall model was significant (*F* = 19.510, *p* < 0.05, *R*^2^ = 0.371, Δ*R*^2^ = 0.337), indicating that the model explained about 37.1% of the variance in personal accomplishment, with personality traits accounting for an additional 33.7% beyond control variables. This represents the highest explanatory power among the three models, underscoring the protective roles of adaptive traits in sustaining secondary PE teachers’ sense of accomplishment. The results are presented in Table 7.

## 4. Discussion

This study investigated the associations between personality traits and burnout among secondary PE teachers, focusing on the subdimensions of emotional exhaustion, depersonalization, and personal accomplishment. Key interpretations are summarized below.

### 4.1. Emotional Exhaustion and Personality Traits

Neuroticism was positively associated with emotional exhaustion, whereas extraversion showed an inverse association. Teachers high in neuroticism, characterized by emotional instability and low stress tolerance, were more prone to exhaustion, whereas extraverted teachers drew on social interactions to bolster their emotional resilience ([2]; [16]; [18]; [45]). The findings align with prior work emphasizing the roles of stress-coping strategies and emotional regulation in teacher burnout ([10]; [37]; [43]). The model explained 22.7% of the variance in emotional exhaustion, indicating that emotional strain among PE teachers is predominantly driven by neuroticism, with extraversion serving as a modest protective factor that mitigates daily stress and emotional depletion.

Beyond general teaching demands, PE teachers face intense physical and safety responsibilities ([30]; [64]). Teachers high in neuroticism may experience constant vigilance about student injury or disciplinary incidents, which accelerates emotional depletion. In contrast, extraverted teachers, who tend to seek social interaction and positive engagement with students, may replenish emotional resources through classroom rapport. Thus, emotional exhaustion in PE teaching reflects not only the workload intensity but also the dispositional capacity to recover from recurring stressors.

### 4.2. Depersonalization and Personality Traits

Depersonalization was significantly associated with extraversion, agreeableness, neuroticism, and openness. Higher extraversion and agreeableness were related to lower depersonalization, indicating that positive relationships with students and colleagues may buffer interpersonal strain. In contrast, higher neuroticism and openness were related to higher depersonalization, suggesting that difficulties with emotional regulation or autonomy-oriented dispositions can sometimes foster emotional distance in structured school settings ([11]; [17]; [27]). Although the explained variance was more modest (*R*^2^ = 0.253), these results underscore that depersonalization arises from the combined influence of multiple traits rather than a single dominant factor.

In the PE context, depersonalization often arises from the sustained emotional demands of managing students and maintaining discipline in physically active and performance-oriented environments ([36]; [52]). Teachers high in neuroticism may struggle with emotional regulation, leading to emotional distancing ([11]), whereas those high in openness may experience frustration when their creative autonomy is constrained by rigid curricula ([17]). Accordingly, depersonalization reflects both the interpersonal fatigue and tension between autonomy-seeking dispositions and institutional constraints.

### 4.3. Personal Accomplishment and Personality Traits

Personal accomplishment was positively associated with extraversion, agreeableness, conscientiousness, and openness, and negatively with neuroticism. Teachers high in adaptive traits tended to report greater self-efficacy and achievement ([8]; [31]), whereas higher neuroticism was linked to lower self-evaluation and helplessness ([56]; [68]). Notably, this model explained the largest variance (*R*^2^ = 0.371), highlighting the protective role of adaptive traits in sustaining professional efficacy and intrinsic motivation ([51]; [53]; [62]).

Within PE instruction, teachers high in extraversion, agreeableness, and conscientiousness are more likely to maintain positive classroom relationships and a clear sense of instructional purpose ([8]; [31]). Moreover, active participation in professional learning communities and peer collaboration enhances reflective practice and job satisfaction among Korean PE teachers ([34]; [70]). Together, these findings indicate that adaptive traits underpin both sustained professional efficacy and motivation in the physically and emotionally demanding environment of PE.

### 4.4. Theoretical Implications

The results support the utility of the Big Five framework for understanding burnout, with distinct traits differentially associated with each dimension. In particular, the robust association of neuroticism with emotional exhaustion is consistent with meta-analytic evidence linking emotional instability to higher strain ([1]; [54]). This association highlights the central role of neuroticism in shaping teachers’ emotional experiences in demanding school contexts. Individuals high in neuroticism are more likely to perceive routine challenges as threatening; to react with heightened emotional sensitivity; and to rely on maladaptive coping strategies, such as avoidance or rumination ([4]; [24]). These tendencies increase stress reactivity and diminish the ability to recover from daily emotional strain, thereby intensifying feelings of exhaustion. Within educational settings, such individuals may experience persistent emotional vigilance, anxiety about classroom management or student outcomes, and difficulty detaching from work-related concerns—all of which accelerate emotional depletion. Recent studies further show that neuroticism is negatively related to self-efficacy and emotion regulation, suggesting that teachers high in neuroticism are more vulnerable to cumulative stress and burnout over time ([11]; [60], [61]). Consequently, neuroticism can be viewed as a dispositional vulnerability factor that amplifies the emotional demands of teaching and contributes to sustained exhaustion under high workload conditions.

In contrast, openness influences burnout through contextual incongruence rather than emotional sensitivity. The positive link between openness and depersonalization suggests a mismatch between autonomy-oriented dispositions and structured institutional contexts. Teachers high in openness value creativity and autonomy, yet PE instruction often adheres to standardized curricula and strict safety protocols. This mismatch can generate frustration and emotional distancing, manifesting as higher depersonalization. Thus, openness may act as a context-dependent trait: adaptive when autonomy is supported, but maladaptive under rigid conditions. This pattern, diverging from Western findings that associate openness with adaptability ([8]; [17]), likely reflects contextual moderators specific to South Korea, where hierarchical norms and limited autonomy prevail ([22]; [58]). These results collectively support a multi-pathway model of burnout: neuroticism influences burnout through emotional reactivity, while openness influences burnout through a person–environment misfit.

### 4.5. Practical Implications

From a preventive perspective, the findings suggest that personality-aware approaches can inform teacher training, workload design, and professional support systems. For teachers high in neuroticism, trait-informed emotional regulation or mindfulness programs can help manage emotional demands consistent with their dispositions ([9]; [10]; [19]). Teachers high in extraversion or conscientiousness may thrive in leadership or collaboration-intensive roles, leveraging interpersonal strengths ([7]; [39]; [47]). Aligning role allocation and support with personality profiles could promote well-being and reduce burnout. Practically, schools could adopt a low-burden, opt-in framework: brief personality-awareness feedback ([8]); voluntary mentoring or coaching matched to individual strengths; and system-level safeguards, such as protected planning time and streamlined administrative workload ([29]). Such trait-informed practices should complement, rather than replace, broader organizational supports, functioning as preventive strategies that align individual dispositions with the contextual realities of PE teaching.

### 4.6. Personality-Informed Professional Learning and Counseling Supports

Integrating personality awareness into counseling and professional learning community (PLC) design can enhance both engagement and sustainability. Personality-informed supports align with self-determination and social–cognitive frameworks, which emphasize autonomy, competence, and relatedness as key motivational resources ([55]; [60], [61]). By recognizing individual differences in emotional regulation and coping, PLCs can cultivate a psychologically safe climate that promotes authentic engagement and reduces emotional suppression, which are both critical for sustainable teaching efficacy. Recent work in PE professional development further shows that well-functioning learning communities strengthen reflective practice, peer collaboration, and professional identity ([33]; [49]).

Empirical studies of school-based learning communities demonstrate that teachers’ learning and coping strategies vary by personality type ([5]; [13]). In practice, conscientious teachers may anchor task-oriented collaboration, extraverted teachers may foster positive group climates, and teachers high in neuroticism may require sustained socio-emotional support ([29]). Evidence from South Korea illustrates that long-term PE teacher communities of practice (CoPs) can enhance both instructional reflection and pupil engagement through trust-based collaboration and shared inquiry ([34]; [70]). Embedding personality-sensitive reflection and mentoring structures into PLCs could amplify their developmental and affective outcomes.

In South Korean schools, hierarchical norms and collective accountability may discourage open discussion of stress or individual differences. Thus, implementing personality-informed PLCs requires a culturally sensitive approach that normalizes individual variability while maintaining group harmony. Sustained trust and professional intimacy are essential for authentic collaboration ([35]; [70]). Anonymized self-assessments, rotating peer-coaching roles, and flexible reflection protocols can balance personal insight with collective trust ([33]; [35]).

For implementation, schools should adopt low-stakes voluntary participation, combining personality-awareness feedback with optional coaching. Embedding reflection cycles (goal setting → observation/peer feedback → debrief) into PLC routines can normalize participation and reduce stigma ([35]). System-level supports, such as protected planning time and recognition of reflective practice, should complement these efforts. In sum, personality-tailored mentoring and collaborative reflection frameworks can strengthen PLC cohesion, foster adaptive coping, and mitigate burnout among PE teachers.

### 4.7. Limitations and Future Research Directions

This study has several limitations. First, the reliance on self-reported data introduces potential social desirability and common-method bias, and the cross-sectional design precludes causal inference; therefore, the findings should be interpreted as associations rather than causal effects. Second, the sample was limited to secondary PE teachers in one metropolitan region, with a pronounced gender imbalance (over 80% male). While this reflects the demographic reality of South Korean PE teaching, it constrains the generalizability to female or more gender-balanced samples. Third, contextual variables, such as workload, classroom climate, and organizational culture, were not included in the analyses. These variables may influence both the expression of personality traits and their associations with burnout.

Future studies should incorporate these factors, adopt longitudinal or multisite designs, and test personality-informed interventions using multilevel or structural equation models to integrate individual and organizational predictors. In addition, longitudinal or cross-lagged (and where feasible, experience-sampling) designs could clarify temporal ordering, while multilevel moderation tests (e.g., school climate or workload norms × traits) could identify conditions under which personality–burnout relationships are amplified or buffered. Finally, micro-intervention trials (e.g., brief emotional regulation or peer coaching modules), randomized or stratified by trait profile, would help determine for whom personality-informed supports are most effective.

## 5. Conclusions

Using hierarchical multiple regression analyses, this study tested three hypotheses regarding the relationships between the Big Five personality traits and burnout dimensions among secondary PE teachers, supplemented by exploratory analyses of background differences. The analyses revealed three consistent patterns. First, neuroticism emerged as the strongest predictor of emotional exhaustion, supporting H2, whereas extraversion demonstrated a significant buffering effect. Second, depersonalization was negatively predicted by extraversion and agreeableness but positively predicted by neuroticism and openness, partially supporting H1–H3 and indicating a possible person–environment misfit effect for openness in structured PE contexts. Third, personal accomplishment was positively predicted by extraversion, agreeableness, conscientiousness, and openness, and negatively predicted by neuroticism, fully supporting H1 and H2.

Background analyses showed significant differences by teaching level and educational background, while gender, work assignment, and marital status showed no meaningful variations. These findings confirm that personality traits play differentiating roles in shaping burnout subdimensions, with neuroticism acting as a core vulnerability factor and adaptive traits (extraversion, agreeableness, conscientiousness) serving as protective resources.

Overall, the results validate the theoretical model proposed in this study and underscore the importance of personality-informed frameworks for burnout prevention and teacher well-being. Integrating personality awareness into teacher education and support programs may enhance emotional resilience, professional satisfaction, and long-term sustainability in PE teaching.

## Figures and Tables

**Table 1 behavsci-15-01499-t001:** Demographic characteristics of participants (*N* = 240).

Category	Variables	Frequency (N)	Percentage (%)
Gender	Male	198	82.5
	Female	42	17.5
Work Assignment	PE-related	95	39.6
	Non-PE-related	145	60.4
Teaching Level	Middle school	147	61.3
	High school	93	38.8
Educational Background	Undergraduate	150	62.5
	Postgraduate or above	90	37.5
Marital Status	Single	58	24.2
	Married	182	75.8
Total		240	100

**Table 2 behavsci-15-01499-t002:** Descriptive statistics of study variables (*N* = 240).

Variable		*N*	Min	Max	M	SD	Skewness	Kurtosis
Burnout	Emotional Exhaustion	240	0.11	5.78	2.26	1.14	0.467	−0.459
	Depersonalization	240	0.00	6.00	1.34	1.14	1.158	1.409
	Personal Accomplishment	240	0.88	6.00	4.01	1.20	−0.492	−0.472
Personality	Extraversion	240	1.75	4.88	3.27	0.63	−0.026	−0.391
	Agreeableness	240	2.22	4.89	3.89	0.49	−0.047	−0.275
	Conscientiousness	240	2.11	5.00	3.83	0.66	−0.157	−0.725
	Neuroticism	240	1.00	4.38	2.54	0.67	0.038	−0.210
	Openness	240	1.10	5.00	3.35	0.69	−0.239	−0.194

**Table 3 behavsci-15-01499-t003:** Pearson correlations between burnout dimensions and Big Five personality traits (*N* = 240).

Variable	1	2	3	4	5	6	7	8
Emotional Exhaustion	1							
Depersonalization	0.659 *	1						
Personal Accomplishment	−0.197 *	−0.354 *	1					
Extraversion	−0.218 *	−0.227 *	0.417 *	1				
Agreeableness	−0.327 *	−0.374 *	0.440 *	0.221 *	1			
Conscientiousness	−0.284 *	−0.317 *	0.463 *	0.307 *	0.576 *	1		
Neuroticism	0.443 *	0.366 *	−0.432 *	−0.303 *	−0.580 *	−0.532 *	1	
Openness	−0.092	−0.078	0.401 *	0.451 *	0.371 *	0.328 *	−0.293 *	1

* *p* < 0.05.

**Table 4 behavsci-15-01499-t004:** Differences in burnout and personality traits by teaching level and educational background (*N* = 240).

Variable	N (%)	Burnout			Personality				
		EE	D	PA	E	A	C	N	O
		Mean (SD)	Mean (SD)	Mean (SD)	Mean (SD)	Mean (SD)	Mean (SD)	Mean (SD)	Mean (SD)
Teaching Level									
Middle School	147 (61.2)	2.25 (1.15)	1.52 (1.22)	3.89 (1.20)	3.24 (0.62)	3.88 (0.49)	3.84 (0.65)	2.59 (0.66)	3.26 (0.70)
High School	93 (38.8)	2.14 (1.11)	1.04 (0.95)	4.20 (1.18)	3.32 (0.64)	3.90 (0.48)	3.82 (0.68)	2.48 (0.69)	3.49 (0.65)
t-Value		1.391	3.233 *	−1.989 *	−1.029	−0.339	0.149	1.237	−2.526 *
Education Background									
Undergraduate	150 (62.5)	2.28 (1.19)	1.44 (1.20)	3.85 (1.21)	3.19 (0.63)	3.84 (0.48)	3.77 (0.67)	2.57 (0.64)	3.21 (0.69)
Postgraduate or Higher	90 (37.5)	2.24 (1.05)	1.17 (1.03)	4.28 (1.14)	3.40 (0.62)	3.98 (0.49)	3.94 (0.61)	2.50 (0.72)	3.58 (0.64)
t-Value		0.306	1.751 *	−2.683 *	−2.471 *	−2.285 *	−1.966 *	0.847	−4.200 *

Note: EE—emotional exhaustion, D—depersonalization, PA—personal accomplishment, E—extraversion, A—agreeableness, C—conscientiousness, N—neuroticism, and O—openness. * *p* < 0.05.

**Table 5 behavsci-15-01499-t005:** Associations between personality traits and emotional exhaustion (*N* = 240).

Variable	B	SE	*β*	95% CI	t (*p*)	*F* (*p*)	*R* ^2^	Δ*R*^2^
(Constant)	2.409	1.006		[0.437, 4.381]	2.394	9.760 *	0.227	0.219
Extraversion	−0.243	0.121	−0.134	[−0.480, −0.006]	−2.007 *			
Agreeableness	−0.300	0.185	−0.128	[−0.662, 0.062]	−1.621			
Conscientiousness	−0.058	0.131	−0.034	[−0.315, 0.199]	−0.445			
Neuroticism	0.583	0.128	0.344	[0.332, 0.834]	4.537 *			
Openness	0.224	0.115	0.136	[−0.002, 0.450]	1.943			

* *p* < 0.05.

**Table 6 behavsci-15-01499-t006:** Associations between personality traits and depersonalization (*N* = 240).

Variable	B	SE	*β*	95% CI	t (*p*)	*F* (*p*)	*R* ^2^	Δ*R*^2^
(Constant)	4.281	0.991		[2.339, 6.223]	4.321	11.250 *	0.253	0.209
Extraversion	−0.309	0.119	−0.170	[−0.542, −0.076]	−2.595 *			
Agreeableness	−0.613	0.182	−0.261	[−0.970, −0.256]	−3.367 *			
Conscientiousness	−0.171	0.129	−0.098	[−0.424, 0.082]	−1.322			
Neuroticism	0.268	0.126	0.158	[0.021, 0.515]	2.119 *			
Openness	0.358	0.114	0.217	[0.135, 0.581]	3.149 *			

* *p* < 0.05.

**Table 7 behavsci-15-01499-t007:** Associations between personality traits and personal accomplishment (*N* = 240).

Variable	B	SE	*β*	95% CI	t (*p*)	*F* (*p*)	*R* ^2^	Δ*R*^2^
(Constant)	−0.657	0.954		[−2.527, 1.213]	−0.689	19.510 *	0.371	0.337
Extraversion	0.413	0.115	0.217	[0.188, 0.638]	3.602 *			
Agreeableness	0.381	0.175	0.155	[0.038, 0.724]	2.176 *			
Conscientiousness	0.352	0.124	0.193	[0.109, 0.595]	2.830 *			
Neuroticism	−0.236	0.122	−0.133	[−0.475, 0.003]	−1.939 *			
Openness	0.209	0.109	0.121	[−0.005, 0.423]	1.913 *			

* *p* < 0.05.

## Data Availability

The data presented in this study are available on request from the corresponding author. The data are not publicly available due to privacy or ethical restrictions.

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
