# Peer review of "Associations Between Big Five Personality Traits and Burnout Among Secondary Physical Education Teachers in South Korea"

_behavsci, 2025, doi:10.3390/bs15111499_

Round 1

Reviewer 1 Report

Comments and Suggestions for Authors
  1. While the paper mentions stressors like "high-risk activity environments" and "extracurricular programs" for PE teachers, the deep-seated mechanism of how these factors interact with personality traits to produce a unique burnout pattern is insufficiently argued. This makes the study appear more like a replication of known relationships within a specific sample, and the depth of its innovative contribution and theoretical value needs enhancement.
  2. The use of the Big Five personality theory and the Maslach Burnout Inventory as the theoretical foundation is appropriate and standard. The problem, however, is that the research only skims the surface of verifying the relationship between traits and burnout dimensions, failing to utilize the theory for a deeper interpretation of the findings. For instance, the counterintuitive finding that "Openness" positively correlates with "Depersonalization" is mentioned in the discussion as potentially due to "person-environment misfit," but it does not delve into the specific contradictions between the core aspects of "Openness" (e.g., seeking novelty, valuing autonomy) and the potentially highly structured, repetitive nature of secondary school PE teaching. This weakens the theoretical explanatory power.
  3. The research questions – "examining the influence of Big Five Personality traits on occupational burnout" and "differences across background characteristics" – are clear. However, their formulation is overly broad, almost a generic template for all similar studies. The questions lack a more context-specific angle tailored to the particular group of PE teachers. For example, it fails to ask more targeted questions like: "Which personality traits effectively buffer the emotional exhaustion arising from PE teachers' identity-related stress (as mentioned in the text)?"
  4. The conclusions are largely based on the data results and are reasonable. However, there is a significant logical leap: jumping directly from "correlational relationships" to "intervention recommendations". The paper concludes by suggesting "emotion-regulation training for teachers high in Neuroticism," which implies a causal assumption that "modifying personality traits can alleviate burnout." Yet, personality traits are relatively stable, and the study itself employs a cross-sectional correlational design incapable of supporting causal inference. Deriving intervention strategies directly from correlational findings lacks methodological rigor.
  5. The discussion section cites previous literature to support its findings, such as referencing Angelini (2023) and Roloff et al. (2022) meta-analyses regarding the roles of Neuroticism and Extraversion. However, the citation style is primarily supportive listing, i.e., "Our finding X is consistent with Scholar Y's study," lacking in-depth dialogue and integration with the literature. For instance, when discussing the unconventional finding of a "positive association between Openness and Depersonalization," it fails to systematically compare and contrast with studies that found Openness to be a protective factor or non-significant, thereby missing the opportunity to explore potential reasons for this discrepancy (e.g., cultural context, occupational specifics). This limits the critical and theory-building potential of the discussion.
  6. Research Methods:
  •  The paper explicitly states that the Cronbach's alpha for the Agreeableness scale was 0.651, below the commonly accepted threshold of 0.70 in psychological research. Although the authors cite Fornell & Larcker (1981) to deem it acceptable, this nonetheless weakens the reliability of the measurements for this trait. This compromises the interpretability of all correlation and regression results involving Agreeableness (e.g., in Table 6, Agreeableness is the strongest negative predictor of Depersonalization).
  • Insufficient Variable Inclusion: The regression models only included personality traits as predictors, failing to control for important background variables. For example, Table 4 shows significant differences in burnout and personality traits based on teaching level and educational background. Not including these as control variables in the regression analyses means the potential confounding effects of these variables on the personality-burnout relationship cannot be ruled out. The reported "pure" personality effects might be biased.
  1. The language is generally fluent and clearly communicates the research content. However, there is room for improvement in word choice and cohesion.
  • Word Choice: There is some repetition and imprecise expression. For instance, phrases like "informative for identifying..." and "may help enhance..." are used repeatedly in the abstract and conclusion, sounding somewhat vague. Some verb usage could be more precise; e.g., "Aligned with prior work" is less idiomatic than "Consistent with prior work" or "This finding supports previous work."
  • Cohesion and Transition: Logical connections between sentences and paragraphs are sometimes awkward. For example, in the introduction, paragraphs often end with a citation and abruptly start a new topic, lacking transitional sentences to guide the reader through the shift in the argument's flow. This makes the writing feel slightly "jumpy" and affects its smoothness.

Author Response

Comment 1: While the paper mentions stressors like "high-risk activity environments" and "extracurricular programs" for PE teachers, the deep-seated mechanism of how these factors interact with personality traits to produce a unique burnout pattern is insufficiently argued. This makes the study appear more like a replication of known relationships within a specific sample, and the depth of its innovative contribution and theoretical value needs enhancement.

Response 1:  The Introduction was revised to clarify how high-risk activity environments and extracurricular responsibilities interact with personality traits to shape burnout patterns. The added paragraph specifies that neuroticism amplifies vigilance and exhaustion, and conscientiousness may lead to over-commitment and depersonalization, while extraversion and agreeableness initially buffer stress but erode under sustained workload. This emphasizes that burnout among PE teachers reflects a context-dependent interaction between personality and occupational ecology.

Comment 2: The use of the Big Five personality theory and the Maslach Burnout Inventory as the theoretical foundation is appropriate and standard. The problem, however, is that the research only skims the surface of verifying the relationship between traits and burnout dimensions, failing to utilize the theory for a deeper interpretation of the findings. For instance, the counterintuitive finding that "Openness" positively correlates with "Depersonalization" is mentioned in the discussion as potentially due to "person-environment misfit," but it does not delve into the specific contradictions between the core aspects of "Openness" (e.g., seeking novelty, valuing autonomy) and the potentially highly structured, repetitive nature of secondary school PE teaching. This weakens the theoretical explanatory power.

Response 2: Section 4.4 (Theoretical Implications) was expanded to provide a deeper theory-driven explanation. The revised text explains that openness—characterized by autonomy, creativity, and novelty seeking—can conflict with the highly structured, standardized nature of PE teaching, producing frustration and detachment. This strengthens the theoretical linkage between personality and burnout.

Comment 3: The research questions – "examining the influence of Big Five Personality traits on occupational burnout" and "differences across background characteristics" – are clear. However, their formulation is overly broad, almost a generic template for all similar studies. The questions lack a more context-specific angle tailored to the particular group of PE teachers. For example, it fails to ask more targeted questions like: "Which personality traits effectively buffer the emotional exhaustion arising from PE teachers' identity-related stress (as mentioned in the text)?"

Response 3: The final paragraph of the Introduction was revised to specify context-tailored questions focusing on (a) how Big Five traits predict burnout subdimensions, (b) which traits buffer emotional exhaustion related to identity stress, and (c) whether these relations differ by background variables.

Comment 4: The conclusions are largely based on the data results and are reasonable. However, there is a significant logical leap: jumping directly from "correlational relationships" to "intervention recommendations". The paper concludes by suggesting "emotion-regulation training for teachers high in Neuroticism," which implies a causal assumption that "modifying personality traits can alleviate burnout." Yet, personality traits are relatively stable, and the study itself employs a cross-sectional correlational design incapable of supporting causal inference. Deriving intervention strategies directly from correlational findings lacks methodological rigor.

Response 4: Section 4.5 (Practical Implications) was reframed from causal “intervention” to preventive “contextual guidance”. The revision clarifies that emotion-regulation and mindfulness programs function as supportive resources aligned with stable personality traits, not attempts to modify them. References (Carrol et al., 2022; Castillo-Gualda et al., 2019; Fabbro et al., 2020) were added.

Comment 5: The discussion section cites previous literature to support its findings, such as referencing Angelini (2023) and Roloff et al. (2022) meta-analyses regarding the roles of Neuroticism and Extraversion. However, the citation style is primarily supportive listing, i.e., "Our finding X is consistent with Scholar Y's study," lacking in-depth dialogue and integration with the literature. For instance, when discussing the unconventional finding of a "positive association between Openness and Depersonalization," it fails to systematically compare and contrast with studies that found Openness to be a protective factor or non-significant, thereby missing the opportunity to explore potential reasons for this discrepancy (e.g., cultural context, occupational specifics). This limits the critical and theory-building potential of the discussion.

Response 5: We strengthened Section 4.4 by systematically comparing our results with studies where openness was protective (Caprara et al., 2011; De Neve et al., 2023) or non-significant (Alarcon et al., 2009; Widiger & Oltmanns, 2017), and contextualized these discrepancies via Korean cultural and institutional factors (hierarchical norms, limited autonomy).

Comment 6: The paper explicitly states that the Cronbach's alpha for the Agreeableness scale was 0.651, below the commonly accepted threshold of 0.70 in psychological research. Although the authors cite Fornell & Larcker (1981) to deem it acceptable, this nonetheless weakens the reliability of the measurements for this trait. This compromises the interpretability of all correlation and regression results involving Agreeableness (e.g., in Table 6, Agreeableness is the strongest negative predictor of Depersonalization).

Response 6: We added clarification citing empirical precedents showing that moderate α values are common and acceptable for the BFI Agreeableness scale (Basim & Begenirbaş, 2012; Husain et al., 2025).

Comment 7: Insufficient Variable Inclusion: The regression models only included personality traits as predictors, failing to control for important background variables. For example, Table 4 shows significant differences in burnout and personality traits based on teaching level and educational background. Not including these as control variables in the regression analyses means the potential confounding effects of these variables on the personality-burnout relationship cannot be ruled out. The reported "pure" personality effects might be biased.

Response 7: Regression analyses were re-run as hierarchical multiple regressions including teaching level and educational background as Step 1 controls. Tables 5–7 and related text were updated accordingly.

Comment 8: The language is generally fluent and clearly communicates the research content. However, there is room for improvement in word choice and cohesion.

Response 8: The entire manuscript was edited for precision, cohesion, and fluency. Repetitive or vague phrases were replaced with concise alternatives; “burnout dimensions” was standardized; transitional sentences were added to improve logical flow, especially in the Introduction and Conclusion.

Reviewer 2 Report

Comments and Suggestions for Authors

This manuscript presents a relevant and well-structured study that examines the relationships between the Big Five personality traits and burnout among secondary physical education teachers in South Korea. The topic is timely and important, contributing to the growing body of research on individual differences associated with burnout in educational settings. The paper is generally well-organized and offers meaningful insights that can inform both research and practice.

Positive aspects:

  • The study addresses an important research problem, focusing on an underexplored population: physical education teachers. This contributes contextual novelty to the literature on teacher burnout.
  • The manuscript is structured logically and coherently. The methodology is solid, utilizing validated measures, appropriate statistical analyses, and clearly presenting the results.
  • The findings are communicated in a straightforward and understandable manner, supported by well-designed tables.
  • Furthermore, the discussion highlights practical implications for teacher well-being and professional development, connecting personality factors to strategies for preventing burnout.

Aspects to improve:

  • Theoretical Framing: While the literature review is solid, the paper would benefit from a clearer explanation of its theoretical novelty. Please elaborate on how the current study advances previous findings or models that connect personality and burnout, beyond its contextual contributions.
  • Measurement Validity: Consider providing additional psychometric information, such as factor loadings and fit indices, to strengthen confidence in construct validity. Clarify how you handle average variance extracted (AVE) values below .50 and briefly justify the adequacy of internal consistency for each dimension.
  • Methodology: The research objectives are presented clearly; however, the study design, hypotheses, and methodological structure are not explicitly stated. The manuscript would benefit from clearly identifying the research design (e.g., cross-sectional correlational), formulating explicit hypotheses derived from the theoretical background, and linking them more directly to the analytic strategy. This clarification would improve transparency and scientific rigor.
  • Sample Characteristics: The significant gender imbalance and regional limitations should be acknowledged as constraints on generalizing the findings. A brief comment on cultural factors that influence the dynamics between personality and burnout could enhance the interpretation.
  • Statistical Reporting: Include standardized regression coefficients (β) and confidence intervals for all regression paths. Ensure that statistical notation is consistent across tables.
  • Interpretation of Findings: Some associations, such as the link between openness and depersonalization, warrant a deeper theoretical discussion to clarify possible mechanisms or contextual moderators.
  • Language and Style: The English is generally clear, but minor improvements to sentence flow, transitions, and the consistency of terminology (e.g., “burnout dimensions” versus “burnout subscales”) would enhance readability.
  • Limitations and Future Directions: Strengthen this section by discussing the potential for common method bias due to the cross-sectional and self-reported nature of the data. Additionally, suggest how future research might address these limitations, such as through longitudinal or multi-source designs.
Comments on the Quality of English Language

The English language is generally clear and understandable. However, minor stylistic and grammatical improvements would enhance the overall fluency and precision of the text. Careful proofreading is recommended to refine sentence flow and ensure consistency in terminology and tense.

Author Response

Comment 1: Theoretical Framing: While the literature review is solid, the paper would benefit from a clearer explanation of its theoretical novelty. Please elaborate on how the current study advances previous findings or models that connect personality and burnout, beyond its contextual contributions.

Response 1: The theoretical contribution of the study has been clarified in the Introduction. We explicitly explain how this study extends previous personality–burnout models by conceptualizing the Big Five traits not as static predictors but as dynamic components interacting across multidimensional burnout processes. Furthermore, the revised text emphasizes the integration of the Big Five framework with the emotional and motivational ecology of PE teaching—a domain characterized by intensive emotional labor and unique job demands. This revision clarifies the study’s theoretical novelty from a multidimensional and context-sensitive perspective.

Comment 2: Measurement Validity: Consider providing additional psychometric information, such as factor loadings and fit indices, to strengthen confidence in construct validity. Clarify how you handle average variance extracted (AVE) values below .50 and briefly justify the adequacy of internal consistency for each dimension.

Response 2: We clarified the psychometric evidence supporting construct validity and reliability in Section 2.2. Although the overall model fit indices were not reported due to marginal fit in some submodels, we provided additional details to demonstrate adequate convergent validity and internal consistency. Specifically, all standardized factor loadings were significant (p < .001) and exceeded .60, and all composite reliability (CR) values were above .70. As several AVE values were slightly below .50, we explicitly justified their acceptability following Fornell and Larcker’s (1981) guideline that convergent validity is adequate when CR exceeds .60 even if AVE is below .50. Cronbach’s α coefficients across subdimensions ranged from .65 to .89, supporting sufficient internal consistency. These details have been incorporated in Sections 2.2.2 and 2.2.3 of the revised manuscript.

Comment 3: Methodology: The research objectives are presented clearly; however, the study design, hypotheses, and methodological structure are not explicitly stated. The manuscript would benefit from clearly identifying the research design (e.g., cross-sectional correlational), formulating explicit hypotheses derived from the theoretical background, and linking them more directly to the analytic strategy. This clarification would improve transparency and scientific rigor.

Response 3: Section 2.1 (Research Design and Hypotheses) now explicitly identifies the research as a cross-sectional correlational design examining the relationships between Big Five traits and three burnout dimensions. Three theoretically grounded hypotheses (H1–H3) were added, including a contextual hypothesis addressing the openness–depersonalization relationship in structured PE settings.

Comment 4: Sample Characteristics: The significant gender imbalance and regional limitations should be acknowledged as constraints on generalizing the findings. A brief comment on cultural factors that influence the dynamics between personality and burnout could enhance the interpretation.

Response 4: In Section 4.7 (Limitations and Future Research Directions), we explicitly acknowledge the gender imbalance (over 80% male) and the regional limitation of the sample, noting their effects on generalizability. We further added discussion of sociocultural factors—collectivistic work norms, hierarchical school culture, and emotional self-regulation expectations—that may influence the personality–burnout relationship.

Comment 5: Statistical Reporting: Include standardized regression coefficients (β) and confidence intervals for all regression paths. Ensure that statistical notation is consistent across tables.

Response 5: Standardized regression coefficients (β) and 95% confidence intervals (CIs) for all regression paths were added in Tables 5–7. Statistical notation and table formatting were also unified across all results to ensure consistency and clarity. These revisions enhance the transparency and interpretability of the reported findings.

Comment 6: Interpretation of Findings: Some associations, such as the link between openness and depersonalization, warrant a deeper theoretical discussion to clarify possible mechanisms or contextual moderators.

Response 6: Section 4.4 (Theoretical Implications) was expanded to provide a deeper interpretation of the openness–depersonalization association. The discussion now links this result to a person–environment misfit mechanism, explaining how openness-driven needs for autonomy and creativity may conflict with the structured, standardized routines of PE teaching, leading to frustration and emotional detachment.

Comment 7: Language and Style: The English is generally clear, but minor improvements to sentence flow, transitions, and the consistency of terminology (e.g., “burnout dimensions” versus “burnout subscales”) would enhance readability.

Response 7: The entire manuscript was reviewed for consistency and fluency. Terminology such as “burnout dimensions” is now used uniformly, and transitions between paragraphs were refined to enhance readability and logical flow.

Comment 8: Limitations and Future Directions: Strengthen this section by discussing the potential for common method bias due to the cross-sectional and self-reported nature of the data. Additionally, suggest how future research might address these limitations, such as through longitudinal or multi-source designs.

Response 8: Section 4.7 was substantially revised to acknowledge common method bias arising from the self-report, cross-sectional design. We now recommend longitudinal and multi-source approaches in future research and reiterate sample-related and cultural factors that may shape burnout dynamics.